# Identification of Candidate Genes Associated with Type-II Sex Pheromone Biosynthesis in the Tea Geometrid (*Ectropis obliqua*) (Lepidoptera: Geometridae)

**DOI:** 10.3390/insects15040276

**Published:** 2024-04-15

**Authors:** Changxia Xu, Nanxia Fu, Xiaoming Cai, Zhaoqun Li, Lei Bian, Chunli Xiu, Zongmao Chen, Long Ma, Zongxiu Luo

**Affiliations:** 1Tea Research Institute, Chinese Academy of Agricultural Sciences, Hangzhou 310008, China; xuchangxia1012@163.com (C.X.); funanxia@tricaas.com (N.F.); cxm_d@tricaas.com (X.C.); zqli@tricaas.com (Z.L.); bianlei@tricaas.com (L.B.); xiuchunli@tricaas.com (C.X.); zmchen2006@163.com (Z.C.); 2Key Laboratory of Biology, Genetics and Breeding of Special Economic Animals and Plants, Ministry of Agriculture and Rural Affairs, Hangzhou 310008, China; 3College of Life Sciences, Jiangxi Science and Technology Normal University, Nanchang 330013, China

**Keywords:** transcriptomic analyses, gene identification, sex pheromone, biosynthesis, RT-qPCR, *Ectropis obliqua*

## Abstract

**Simple Summary:**

The tea geometrid (*Ectropis obliqua* (Prout)) is a major defoliator of tea plants and seriously affects the tea quality and yield in China. The females release three Type-II sex pheromone components for mate communication, but knowledge about the biosynthesis of these compounds is still limited. In this study, we screened for the candidate genes in the sex pheromone biosynthetic pathway with a combination of comparative RNAseq, phylogenetic relationship, and tissue expression pattern analyses. A total of seven tentative biosynthetic enzymes were identified, including two ELOs (ELO3 and ELO5), two FARs (FAR2 and FAR9), one DEC (CYP4G173), one LIP (LIP1), and one epoxidase (CYP340BD1). Overall, our results provide a foundation for further functional elucidation of the vital genes involved in *E. obliqua* sex pheromone biosynthesis.

**Abstract:**

*Ectropis obliqua*, a notorious tea pest, produces a Type-II sex pheromone blend for mate communication. This blend contains (*Z,Z,Z*)-3,6,9-octadecatriene, (*Z,Z*)-3,9-*cis*-6,7-epoxy-octadecadiene, and (*Z,Z*)-3,9-*cis*-6,7-epoxy-nonadecadiene. To elucidate the genes related to the biosynthesis of these sex pheromone components, transcriptome sequencing of the female *E. obliqua* pheromone gland and the abdomen without pheromone gland was performed. Comparative RNAseq analyses identified 52 putative genes, including 7 fatty acyl-CoA elongases (ELOs), 9 fatty acyl-CoA reductases (FARs), 1 decarbonylase (DEC), 3 lipophorins (LIPs), and 32 cytochrome P450 enzymes (CYPs). Tissue expression profiles revealed that two ELOs (ELO3 and ELO5), two FARs (FAR2 and FAR9), one DEC (CYP4G173), and one LIP (LIP1) displayed either abdomen-centric or -specific expression, suggesting potential roles in sex pheromone biosynthesis within the oenocytes of *E. obliqua*. Furthermore, the tissue expression patterns, combined with phylogenetic analysis, showed that CYP340BD1, which was expressed specifically and predominantly only in the pheromone gland, was clustered with the previously reported epoxidases, highlighting its potential role in the epoxidation of the unsaturated polytriene sex pheromone components. Collectively, our research provides valuable insights into the genes linked to sex pheromone biosynthesis.

## 1. Introduction

Sex pheromones play crucial roles in mediating mating communication and ensuring reproductive isolation among lepidopteran moths. Sex pheromones are usually secreted by female moths from the pheromone gland, which is typically located between the eighth and ninth abdominal segments [1]. To date, the sex pheromones or attractants of over 1600 moth species have been chemically identified [2]. Based on their chemical structures, moth sex pheromones are categorized into three primary types: Type I, which constitutes 75% of moth sex pheromones; Type II, representing 15%; and a miscellaneous type comprising the remaining 10% [3,4]. Type I pheromones are composed of unsaturated straight-chain C10–C18 compounds, which can take the form of alcohols, aldehydes, or acetate esters. Type-II pheromones, primarily found in the Geometridae and Erebidae moth families, are characterized by C17–C23 unsaturated hydrocarbons with 2–3 *cis* double bonds alongside their corresponding epoxides. And the double bonds are typically situated at the 3-, 6-, and 9-carbon positions. Pheromones that do not fit within these two classifications fall into the miscellaneous category. 

The biosynthesis of moth sex pheromones, especially the Type-I variants, has been rigorously explored both chemically and molecularly. Generally, the C10–C18 unsaturated acyclic aliphatic compounds constituting the Type-I sex pheromones are synthesized de novo within the pheromone gland. This synthesis process involves a series of sequential enzymatic reactions, such as desaturation, chain-shortening, reduction, acetylation, and oxidation [5]. Instead of de novo synthesis, isotopic labeling experiments proved that Type-II moth sex pheromones in both winter moths and the fall webworm (*Hyphantria cunea*) are biosynthetically derived from dietary α-linoleic acid (*Z*9,*Z*12-18:COOH) or linolenic acid (*Z*9,*Z*12,*Z*15-18:COOH) [6,7,8]. Oenocyte cells, which are associated with either epidermal cells or fat body cells of the abdomen, are reported to be the biosynthetic sites for those polyene hydrocarbons [9]. Furthermore, in the winter moth (*Erannis bajaria*), the even-numbered pheromone component (3*Z*,6*Z*,9*Z*)-3,6,9-octadecatriene (3*Z*,6*Z*,9*Z*-18:H) was shown to be produced from the addition of malonate to the precursor linolenic acid, followed by α-oxidation, reduction, and decarbonylation of the carboxyl group, and the odd-numbered pheromone component (3*Z*,6*Z*,9*Z*)-3,6,9-nonadecatriene (3*Z*,6*Z*,9*Z*-19:H) was shown to emerge from the reductive decarbonylation of the even-numbered acyl precursors [10]. This biosynthetic procedure is similar to that of the insect cuticular hydrocarbon [11,12]. Therefore, the enzymes involved in the biosynthesis of polyene hydrocarbons are supposed to be fatty acyl-CoA elongases (ELOs), oxidases, fatty acyl-CoA reductases (FARs), and decarbonylases (DECs). Once synthesized, these unsaturated lipophilic precursors are transported to the female pheromone gland via the hemolymph by the lipid transport protein lipophorin (LIP) [13,14]. In the pheromone gland, these polyene hydrocarbons are either stored, released unaltered, or epoxidized prior to release if the oxygenated epoxides are pheromone components. Cytochrome P450 enzymes (CYPs), which exhibit high expression levels in female pheromone glands, are reported to be the epoxidases. Specifically, a CYP from the CYP340 family in the Japanese giant looper (*Ascotis selenaria*) was proven to be the *Z*3-specific epoxidase that produces the sex pheromone component *cis*-3,4-epoxy-(6*Z*,9*Z*)-6,9-nonadecadiene (epo3,*Z*6,*Z*9-19:H) [15]. Additionally, CYP341B14 was identified in both *H. cunea* and the mulberry tiger moth (*Lemyra imparili*) as the *Z*9-specific epoxidase that generates *cis*-9,10-epoxy-(3*Z*,6*Z*)-3,6-henicosadiene (*Z*3,*Z*6,epo9-21:H) and *cis*-9,10-epoxy-(3*Z*,6*Z*)-3,6-tricosadiene (*Z*3,*Z*6,epo9-23:H) [16,17]. However, other enzymes integral to the Type-II sex pheromone biosynthetic pathway remain elusive. 

The tea geometrid (*Ectropis obliqua*) stands as a significant leaf-chewing pest at Chinese tea plantations, inflicting considerable economic damage. The female sex pheromone of *E. obliqua* comprises a blend of *Z*3,*Z*6,*Z*9-18:H, (*Z,Z*)-3,9-*cis*-6,7-epoxy-octadecadiene (*Z*3,epo6,*Z*9-18:H), and (*Z,Z*)-3,9-*cis*-6,7-epoxy-nonadecadiene (*Z*3,epo6,*Z*9-19:H) [18]. This composition is the same as other documented geometrid moths whose sex pheromones predominantly feature Type-II components, which include polyene hydrocarbons and their corresponding epoxides. Given this pattern, it is hypothesized that the biosynthesis of *Z*3,*Z*6,*Z*9-18:H and 3*Z*,6*Z*,9*Z*-19:H in *E. obliqua* mirrors that of the winter moth (*E. bajaria*). Namely, the female tea geometrid utilizes dietary *Z*9,*Z*12,*Z*15-18:COOH as a precursor that is subjected to chain elongation, α-oxidation, reduction, and decarboxylation within oenocyte cells to generate polyene hydrocarbons. Similar to the moths utilizing *Z*3- and *Z*9-specific epoxides as sex pheromones, the unsaturated *Z*3,*Z*6,*Z*9-18:H and 3*Z*,6*Z*,9*Z*-19:H are subsequently conveyed via hemolymph to the pheromone gland for epoxidation, producing the two corresponding *Z*6-specific epoxides. Despite these insights, the biosynthetic enzymes that are critical to the sex pheromone production of *E. obliqua* have yet to be characterized.

In the present study, we focused on the biosynthesis of the sex pheromone in female *E. obliqua*. Initially, we constructed transcriptome databases for the female pheromone gland and the abdomen excluding the pheromone gland (hereafter referred to as the “abdomen”). Through comparative transcriptomic analyses, we pinpointed potential biosynthesis-related candidate genes that are predominantly present in the abdomen and pheromone gland. Subsequently, we scrutinized the phylogenetic relationships of the identified FARs and CYPs. Additionally, the tissue-specific expression of these candidate genes was further investigated using real-time quantitative PCR (RT-qPCR). Our findings can serve as a foundation for the functional characterization of the vital genes that are involved in sex pheromone biosynthesis in the tea geometrid (*E. obliqua*).

## 2. Materials and Methods

### 2.1. Insect Rearing and Tissue Collection

Larvae of *E. obliqua* were collected from the experimental field of the Tea Research Institute, Chinese Academy of Agricultural Sciences in Hangzhou, Zhejiang Province, China (30.10° N, 120.5° E). Pupae were sexed, and emerging adults were collected daily and reared separately with a 10% honey water solution in a climate chamber. The chamber was controlled to maintain a temperature of 25 ± 1 °C, a relative humidity of 70 ± 5%, and a light cycle of 14 L:10 D. For transcriptome sequencing, the pheromone glands and abdomens of the virgin female *E. obliqua* were dissected 2–3 days post-eclosion during the scotophase, which is the peak calling period. To collect the rod-shaped pheromone gland, the terminal parts of the abdomen of a virgin female were gently squeezed with tweezers. Then, the extruded pheromone gland was excised with a pair of fine scissors. Fifty pheromone glands were pooled and taken as one biological replicate, and a single abdomen was considered a biological replicate. For RT-qPCR analysis, various tissues, including the head, thorax, antenna, leg, abdomen, and pheromone gland, were dissected from female moths between 2 and 3 days post-eclosion during the scotophase. And tissues collected from five female moths were pooled as one biological replicate. For each type of tissue sample, three biological replicates were used. All excised tissues were immediately transferred to liquid nitrogen and subsequently stored at −80 °C for later experiments.

### 2.2. cDNA Library Construction and RNA Sequencing

Total RNA was extracted using TRIzol^®^ reagent (Invitrogen, Carlsbad, CA, USA) according to the manufacturer’s instructions. Genomic DNA was removed using DNase I (Takara Bio, Beijing, China). RNA quality was assessed using an Agilent 2100 Bioanalyzer (Agilent, Beijing, China), and quantification was performed with NanoDrop™ 2000/2000c Spectrophotometers (Thermo Fisher Scientific, Waltham, MA, USA). Approximately 1 μg of high-quality RNA was utilized to construct a cDNA library for Illumina HiSeq sequencing (Illumina Inc., San Diego, CA, USA) using a HiSeq PE150 platform (Mingkebio, Hangzhou, China) in accordance with the manufacturer’s guidelines. These methods and the analysis of the transcriptome sequencing results were performed utilizing the standardized approaches described by Zhang et al. [19]. 

### 2.3. De Novo Assembly and Annotation

To exclude adapters and low-quality reads, the raw paired-end reads underwent trimming and quality control using Trimmomatic version 0.40 (http://www.usadellab.org/cms/?page=trimmomatic, accessed on 5 May 2023) with the default settings. Trinity version 2.15.1 (https://github.com/trinityrnaseq/trinityrnaseq/releases, accessed on 6 May 2023) was employed for de novo transcriptome assembly with the default options [20]. The assembled transcripts were then aligned and annotated using BLASTX against various protein databases, namely the non-redundant database (Nr), Clusters of Orthologous Groups (COG), Swiss-Prot (SWSS), and Kyoto Encyclopedia of Genes and Genomes (KEGG), with a typical E-value cutoff of <1.0 × 10^−5^. The BLASTX outputs from the Nr database were further subjected to GO annotation using Blast2GO (http://www.blast2go.com/b2ghome, accessed on 8 May 2023).

### 2.4. Transcript Expression Level Quantification

The expression levels of the identified transcripts were calculated with the software RSEM v1.3.3 (http://deweylab.github.io/RSEM/, accessed on 10 May 2023). The FPKM method (Fragments Per Kilobase of transcript per Million mapped reads) was applied for the analysis using the following formula: FPKM(A) = 10^6^ C/(M × L/10^3^), where FPKM (A) represents the expression abundance of a transcript; C (cDNA fragments) is the number of fragments uniquely aligned to a transcript; M (mapped fragments) refers to the total number of fragments uniquely aligned to all identified transcripts; and L stands for the length of the transcript (kb). The Bonferroni t-test was used to analyze the difference in FPKM between the abdomen and the pheromone glands.

### 2.5. Sequence Analysis and Phylogenetic Analysis

The putative enzymes linked to *E. obliqua* sex pheromone biosynthesis (namely ELO, FAR, DEC, LIP, and P450 epoxidase) were identified from the NR annotation results by searching the enzymes’ names. All retrieved sequences were validated using the web-based BLASTX tool available on the NCBI-BLAST network server (http://blast.ncbi.nlm.nih.gov/, accessed on 15 May 2023). The open reading frames (ORFs) of these putative genes were predicted utilizing the ORF finder tool (https://www.ncbi.nlm.nih.gov/orffinder accessed on 15 May 2023).

To further assign the candidate genes putative functions in the *E. obliqua* pheromone biosynthetic pathways, genes encoding FAR and P450 epoxidase underwent phylogenetic analysis. This process was carried out based on the amino acid sequences they encoded in comparison with the sequences documented in other insects. Specifically, the FAR dataset comprised 8 sequences from *E. obliqua* and 39 from other lepidopteran moths. Meanwhile, the CYP dataset contained 24 sequences from *E. obliqua* and 59 from other lepidopteran species. The alignment of these candidate genes with sequences from other species was executed using ClustalW, and phylogenetic trees were constructed with MEGA 7.0 [21]. Specifically, the tree for FARs was generated using the neighbor-joining method, incorporating the p-distance model, while the maximum-likelihood method based on the Jones–Taylor–Thornton (JTT) model was employed for the phylogenetic analysis of CYP. For both analyses, a bootstrap value of 1000 was used, and the default settings were used for all other parameters. The amino acid sequences harnessed for the phylogenetic analysis of FARs and CYPs are provided in Appendix A, respectively.

### 2.6. RT-qPCR Analyses

For the RT-qPCR analysis, 1 μg of total RNA from various tissues (head, thorax, leg, antenna, abdomen, and pheromone gland) was reverse-transcribed into cDNA using a HiScript^®^ Q RT SuperMix for qPCR (+gDNA wiper) kit (Vazyme, Nanjing, China). The qPCR procedures were conducted using a Roche LightCycler 480 (Stratagene, La Jolla, CA, USA) using the protocol described by Wang et al. [22]. Each sample had three technical replicates drawn from three biological replicates. Any technical replicate with a Cq difference greater than 0.5 was omitted. For normalization, glyceraldehyde-3-phosphate dehydrogenase and β-actin were selected as housekeeping genes [23]. Primers were designed using primer3-plus (https://www.primer3plus.com/, accessed on 18 May 2023), and their sequences are documented in Appendix A. All procedures were performed according to the MIQE guidelines [24]. Relative gene expression levels were determined using the comparative 2^−∆∆CT^ method [25]. To analyze the differences in gene expression levels among multiple tissues, a one-way analyses of variance (ANOVA) was performed with Tukey’s post hoc test. The level of significance was set at *p* < 0.05. Figures were constructed with GraphPad Prism version 8.0 (GraphPad Software Inc., La Jolla, CA, USA).

## 3. Results

### 3.1. Transcriptomic Sequencing and Functional Annotation of Unigenes

The pheromone glands and abdomens of female *E. obliqua* were sequenced, generating 46,202 unigenes with an average length of 1039 bp after filtering out redundant and low-quality sequences (Table 1). Of these unigenes, 17,683 (38.27%), 7499 (16.23%), 13,283 (28.75%), 7977 (17.27%), and 9435 (20.42%) were annotated in NR, GO, COG, KEGG, and SWSS, respectively. These unigenes were categorized into three functional groups based on their GO annotations: biological processes, cellular components, and molecular functions. Based on sequence homology, 7499 unigenes (16.23%) from *E. obliqua* received annotations. And the annotated genes were primarily enriched for single-organism cellular processes (5107), followed by organic substance metabolic processes (4077) in the biological process classification (see Appendix A). For the cellular component and molecular function categories, the unigenes were predominantly represented in cell parts (5049) and protein binding (1428), respectively. In addition, KEGG analyses revealed that the unigenes were assigned to 322 different pathways, where metabolic pathways were dominant (1256 unigenes), followed by the biosynthesis of secondary metabolites, which involved 433 unigenes (Appendix A).

### 3.2. Identification of Putative Genes Related to Sex Pheromone Biosynthesis 

Originating from linolenic acid, the biosynthesis of the polyene hydrocarbon sex pheromone components in *E. obliqua* is supposed to proceed with consecutive reactions, including chain elongation, oxidation, reduction, and decarbonylation, within the oenocytes of the abdomen. Since no enzymes involved in α-oxidation for sex pheromone biosynthesis have been reported in the literature, we only looked for ELOs, FARs, DECs, and LIPs in the female abdomen transcriptome. Within the database of the *E. obliqua* transcriptome, we identified 18 sequences bearing resemblance to the ELOs of other lepidopteran insects available in the NCBI GenBank (Appendix A). Notably, seven of the ELOs exhibited higher FPKM values compared to the pheromone gland, four of which were expressed significantly more in the abdomen (Figure 1A). Furthermore, our analyses pinpointed 22 FAR-encoding genes, 2 DEC-encoding genes, and 3 LIP-encoding genes (Appendix A). Among these, nine FARs, one DEC, and three LIPs were posited to play roles in the sex pheromone biosynthesis of *E. obliqua*, given their elevated FPKM values in the abdomen relative to the pheromone gland (Figure 1A).

For moths utilizing alkenyl sex pheromones, the optional epoxidation of alkenes (C3, C6, or C9) confers further diversity to the chemical structures of pheromone components [15]. The CYPs, which are highly expressed specifically in the female moth pheromone gland, were reported to catalyze this epoxidation [15,16,17]. From the database of the *E. obliqua* sex pheromone gland transcriptome, a total of 76 CYPs were identified (Appendix A). According to the standard nomenclature, these CYPs were categorized into four primary clans: a mitochondrial clan (11 genes), a CYP2 clan (3 genes), a CYP3 clan (37 genes), and a CYP4 clan (25 genes) (Appendix A). Notably, a total of 48 CYPs exhibited higher FPKM values in the pheromone gland, and 32 of them were statistically significantly abundant, suggesting potential roles in pheromone gland metabolism (Figure 1B). 

### 3.3. Phylogenetic Analyses

Phylogenetic relationship analyses was applied to narrow down the list of candidate FARs. FAR6 was omitted from the phylogenetic analysis because its amino acid length is less than 130. The phylogenetic relationships of the eight remaining FARs are depicted in Figure 2. Specifically, FAR2, FAR5, and FAR9 clustered with orthologs from other lepidopteran species in the same clade, while FAR7 established a distinct clade with SiFAR QLI62003. FAR1, FAR3, and FAR4 collectively formed another cluster. Importantly, only FAR8 was classified within the lepidopteran pgFAR clade, implying potential functional similarities among these genes (Figure 2).

To elucidate the phylogenetic relationships among the CYPs primarily expressed for sex pheromone biosynthesis, a phylogenetic tree was constructed using the maximum-likelihood method in MEGA 7.0 (Figure 3). Of the 32 CYPs with significant FPKM values, we selected 24 for this analysis. Eight CYP candidates were excluded due to their amino acid lengths being less than 100.

As described in Figure 3, the 24 pheromone-specific CYPs were grouped into four distinct clans: CYP2 (brown), CYP3 (purple), CYP4 (light blue), and mitochondrial CYP (green). And most of them were clustered in the CYP3 and CYP4 clans. Given that previously identified epoxidases (Asepo1, Liepo1, and HcunCYP341B14) belong to the CYP4 clan, our analyses focused on the CYPs within this clan. Specifically, seven CYPs clustered alongside their respective orthologs from other lepidopteran CYPs into various families within the CYP4 clan. These included four CYPs (CYP4L71, CYP4L44-N-term, CYP4AU1, and CYP4AU2) in the CYP4 family, one CYP (CYP367A1-N-term) in the CYP367 family, one CYP (CYP340BD1) in the CYP340 family, and one CYP (CYP341U1) in the CYP341 family. Interestingly, CYP340BD1 and CYP341U1, which demonstrated strong expression patterns specific to the pheromone gland (Figure 1B, were aligned within the same clade as 3,4-epoxidase and the 9,10-epoxidases, respectively. This finding suggests their potential roles in the 6,7-epoxidation of the unsaturated sex pheromone component *Z*3,*Z*6,*Z*9-18:H in the tea geometrid (*E. obliqua*).

### 3.4. Tissue Expression Profiles of Putative Genes Involved in E. obliqua Sex Pheromone Biosynthesis

To further delineate the genes involved in the sex pheromone biosynthesis of *E. obliqua*, we investigated the tissue expression profiles of the identified genes across various female tissues, including the head, thorax, abdomen, legs, antennae, and pheromone glands, using RT-qPCR.
Fatty acyl-CoA elongase

The initial step in the sex pheromone biosynthetic pathway of *E. obliqua* is chain elongation. In brief, linolenic acid undergoes elongation, presumably through the action of the microsomal ELO enzyme. This enzyme extends the unsaturated acyl chain by two carbons by undergoing a condensation reaction with malonyl-CoA, leading to the formation of the C20 alkatriene.

Analyses of the tissue expression profiles revealed that none of the seven ELOs displayed a high, abdomen-specific expression pattern. Notably, ELO1 was the only ELO with significantly elevated expression in the antennae, while ELO2 had higher expression levels in the antennae, head, and legs compared to the pheromone glands, abdomen, and thorax. ELO4 transcripts were abundantly detected in the legs, pheromone gland, and antennae of *E. obliqua*. Conversely, the other four candidate ELOs (ELO3, ELO5, ELO6, and ELO7) displayed predominant expression in adult female legs. Nevertheless, among the seven ELO candidates, ELO3 and ELO5 exhibited higher expression levels in the abdomen of *E. obliqua* compared to the antennae, pheromone gland, and thorax. These findings suggest that these two ELOs may play pivotal roles in elongating the linolenic acid chain for sex pheromone biosynthesis (Figure 4A).
Fatty acyl-CoA reductase

According to the RT-qPCR results, six of the nine FARs (FAR1, FAR3, FAR4, FAR5, FAR7, and FAR8) displayed antennae-biased expression patterns. In contrast, FAR6 showed the most pronounced expression in the head (Figure 4B). Notably, while both FAR2 and FAR9 exhibited high expression levels in the abdomen, only FAR9 was expressed significantly more in the abdomen compared to the other tissues. This finding suggests that FAR9 may play a role in the production of *Z*10,*Z*13,*Z*10-19:Ald in the *E. obliqua* sex pheromone biosynthetic pathway.
P450 decarbonylase

Insects employ an aerobic mechanism involving CYP4G for the oxidative decarbonylation of aldehydes, generating hydrocarbons [26]. Consequently, it is postulated that the unsaturated hydrocarbon sex pheromone in *E. obliqua* (*Z*3,*Z*6,*Z*9-18: H) is derived from the decarbonylation of the aldehyde *Z*10,*Z*13,*Z*16-19: Ald. Analyses of tissue expression profiles demonstrated that CYP4G173 exhibited significant high expression in the abdomens of female *E. obliqua* compared to other tissues (Figure 4C). This is in consistent with the RNAseq results, suggesting that CYP4G173 is likely to act as a decarbonylase in *E. obliqua* sex pheromone biosynthesis.
Lipophorin

From the database of the female *E. obliqua* abdomen transcriptome, three putative LIPs exhibiting abdomen-biased expression profiles were identified (Figure 1). The RT-qPCR results indicated that LIP1 exhibited the highest expression level in the abdomen. Meanwhile, LIP2 and LIP3 were predominantly expressed in the thorax and antenna, respectively (Figure 4D). These findings suggest that LIP1 may play a pivotal role in transferring the unsaturated hydrocarbon sex pheromones produced in oenocytes.
Epoxidase


The tissue-specific expression patterns of 32 CYPs were examined in six different tissues of female *E. obliqua* moths using RT-qPCR. According to their tissue expression patterns, these CYPs could be divided into four groups (Figure 5). As shown in Figure 5A, five CYPs (CYP340BD1, CYP4L71, CYP4AU1, CYP6B247, and CYP333A19-C-term) exhibited significantly elevated expression levels in the female pheromone gland compared to the five other tissues. Additionally, CYP4L44-N-term was expressed with significant high levels in the pheromone gland, head, and legs, and CYP332A1 and CYP4AU2 were observed to be upregulated in both the pheromone gland and thorax compared to the other tissues. Notably, among these eight CYPs, CYP340BD1 and CYP4L71 were the most abundantly expressed in the pheromone gland. The pheromone gland-specific expression patterns, combined with the dominant high expression levels of these two CYPs, underscore their potential roles in epoxidizing the unsaturated hydrocarbon sex pheromones in *E. obliqua*.

Furthermore, among the 32 preselected CYPs, seven demonstrated pronounced accumulated expression patterns specifically in the antennae, and five additional CYPs also exhibited antennae-preferential expression patterns (Figure 5B), while six CYPs were preferentially expressed in the head. Moreover, significant high levels of CYP18A1 and CYP305B1 expression were observed in the head (Figure 5C). In addition, five CYPs were detected to be preferentially expressed in the legs, and CYP4G174-fragment1 and CYP341U1 were found to be significantly upregulated in the legs (Figure 5D). CYP6AB14-fragment5 was predominantly expressed in the thorax and antennae (Figure 5D). In terms of its relative abundance, CYP4G174-fragment1 was abundantly expressed in all test tissues, with a statistically upregulated pattern in the legs compared to the other tissues. And none of the 32 CYPs displayed enhanced abundance in the abdomen.

## 4. Discussion

With the advent of next-generation sequencing, RNA sequencing has become a powerful tool for analyzing the transcriptomes of a plethora of moth pheromone glands. So far, the transcriptomes of pheromone glands from over 25 moth species across 10 families have been sequenced and characterized [27]. Notably, the majority have been from moths producing Type-I sex pheromones, which are de novo synthesized in the pheromone glands. In contrast, the polyene hydrocarbon components of Type-II sex pheromones are first synthesized in oenocytes located in the female abdomen and subsequently are transferred to the pheromone gland for final modification or release [28].

In this study, to uncover all genes involved in Type-II sex pheromone biosynthesis in *E. obliqua*, we generated transcriptome databases from the pheromone glands and from the abdomens of 2- to 3-day-old female moths, which is the calling period of this moth [29]. Through comparative transcriptomic analysis, we identified a total of 114 genes presumably involved in Type-II sex pheromone biosynthesis. Our results represent the first comprehensive screening for genes associated with Type-II sex pheromone biosynthesis. Our work lays a basis for further understanding sex pheromone biosynthesis at the molecular level in *E. obliqua* and other lepidopteran insects.

Due to its long-chain features and fatty acid origin, the primary enzymatic steps of Type-II sex pheromone biosynthesis proceed similarly to those of insects’ cuticle hydrocarbon synthesis. For example, several involved enzymes, including ELOs, FARs, DECs (CYP4Gs), and LIPs, are commonly needed for both biosynthetic pathways [26,30,31]. Compared to FARs, little attention has been paid to ELOs, DECs, and LIPs. Most ELOs and DECs (CYP4Gs) associated with cuticular hydrocarbon biosynthesis have been well characterized [32]. We speculated that the tea geometrid (*E. obliqua*) employs analogous enzymes to synthesize the unsaturated compounds *Z*3,*Z*6,*Z*9-18:H and *Z*3,*Z*6,*Z*9-19:H. Within the abdomen transcriptome of *E. obliqua*, seven ELOs and one DEC exhibited abdomen-specific expression patterns. Further tissue-specific expression analysis showed that among the seven identified ELOs, ELO3 and ELO5 were more prominently expressed in the abdomen of *E. obliqua* than in the antennae, pheromone gland, and thorax. This finding suggests that these two ELOs may play a role in converting linolenic acid to *Z*11,*Z*14,*Z*17-20:CoA. Regarding the two identified DECs, only CYP4G173 was significantly highly expressed in the abdomens of female *E. obliqua*, implying a potential role in decarbonylating aldehyde intermediates to produce 3,6,9-triene. Previous research suggested that the trienyl precursor binds with high-density LIP and is transported via the hemolymph to the pheromone glands for epoxidation [14]. Similarly, we identified three LIPs in the abdomen of *E. obliqua*, with LIP1, due to its high expression level in the abdomen, serving as the most probable candidate for transferring trienyl precursors.

FARs catalyze the reduction of fatty acyl-CoA to its corresponding alcohol or aldehyde in insect pheromone biosynthesis and play an important role in determining the proportion of each component in the pheromone bouquet [33,34]. In moths, an increasing number of FAR genes dedicated to Type-I sex pheromone biosynthesis have been functionally characterized across various lepidopteran species [35,36,37]. These FARs cluster into a unique group called pgFARs, which exhibit unique specificity and selectivity for fatty acyl substrates [35]. In the abdomen transcriptome of *E. obliqua*, nine FARs were identified with higher expression in the abdomen than in the pheromone gland. Only FAR8 aligned with the lepidopteran pgFAR clade, implying its functionalities are similar to those of pgFARs. However, previous studies have shown that pseudogenes can be produced via gene duplication during moth sex pheromone evolution [38]. Functional characterization is necessary to check whether FAR8 still has a function in *E. obliqua* pheromone glands. Interestingly, FAR8 was shown to display predominant expression in the antennae. This led to the proposal that FAR8 converts specific unsaturated fatty acids into corresponding alcohols or aldehydes during antennae metabolism. Since the production of the aldehyde precursors proceeds in oenocytes that are adhered to the abdomen integument, the FARs pertinent to *E. obliqua* sex pheromone synthesis should mainly be enriched in the abdomen. Tissue expression profiles have shown that only FAR9 has statistically higher expression in the abdomen compared to other tissues. This result suggests its probable involvement in producing aldehyde precursors in the sex pheromone biosynthetic pathway of *E. obliqua*. To substantiate this hypothesis, a more comprehensive functional analysis is warranted.

Contrasting with Type-I sex pheromone and cuticle hydrocarbon biosynthesis, two pivotal biosynthetic steps play crucial roles in determining the unique species-specific sex pheromone components in *E. obliqua*. The first key and rate-limiting reaction is the α-oxidation of *Z*11,*Z*14,*Z*17-20:CoA, resulting in a loss of one carbon to give *Z*10,*Z*13,*Z*16-19:CoA. Although the α-oxidation of myristic acid has been shown to be catalyzed by P450-like enzymes in the bacterium *Sphingomonas paucimobilis* [39], the α-oxidation of unbranched fatty acids in insects has not been reported before. As a result, it is difficult to identify potential enzymes responsible for the α-oxidation reaction within the sex pheromone biosynthetic pathway in *E. obliqua*. Nonetheless, pinpointing enzyme(s) producing *Z*10,*Z*13,*Z*16-19:CoA remains crucial, as it would significantly enrich our understanding of the intricate mechanisms behind sex pheromone biosynthesis and evolution.

The second critical step entails the selective epoxidation of unsaturated hydrocarbon precursors in the pheromone gland. Previous studies have shown that CYPs from different families have unique roles in the region-specific (*Z*3 and *Z*9) epoxidation of alkenyl precursors in moths with Type-II sex pheromones. Specifically, As_epo1 (CYP340BD2) from the CYP340 family epoxidizes the *Z*3 double bond, whereas Hc_epo1 (CYP341B14) and Li_epo1 (CYP341B14) from the CYP341 family are the *Z*9 double bond-specific epoxidases [17]. Intriguingly, while the amino acid sequence of As_epo1 shares a mere 9.5% similarity with either Hc_epo1 or Li_epo1, all three enzymes are P450 members of clan 4 and exhibit high expressional level specifically in the pheromone glands [15]. Therefore, for the tea geometrid (*E. obliqua*), it is hypothesized that clan 4 CYPs with an affinity for the pheromone glands are responsible for the epoxidation of the *Z*6 double bonds in unsaturated hydrocarbons. Through comparative RNAseq analyses, 32 CYPs were found to be more prominently expressed in the pheromone gland of *E. obliqua* than in the abdomen based on FPKM values (Figure 1). Analyses of tissue expression profiles revealed that five CYPs (CYP340BD1, CYP4L71, CYP4AU1, CYP6B247, and CYP333A19-C-term) were significantly upregulated in the female pheromone gland compared to the other tissues. Furthermore, CYP340BD1 and CYP4L71 were the two most dominant CYPs in the pheromone gland. Phylogenetically, CYP340BD1 closely aligned with the *Z*3 double bond-specific epoxidase As_epo1, whereas CYP4L71 diverged into a separate clade that was distinct from the CYP340 and CYP341 families to which the known epoxidases belong. In conclusion, both tissue expression patterns and phylogenetic data suggest that CYP340BD1 is likely to play a role in the epoxidation of unsaturated polyene sex pheromone components. Functional analyses of CYP340BD1 is essential and will ultimately help in the identification of the *Z*6 double bond-specific epoxidase. Notably, *E. obliqua* uses both *Z*3,epo6,*Z*9-18:H and *Z*3,epo6,*Z*9-19:H as sex pheromone components. Previous enzyme assays have demonstrated that epoxidases exhibit low substrate specificity concerning the aliphatic carbon chain length. Therefore, further studies are needed to test whether *E. obliqua* recruits one CYP or two independent CYPs to produce the two epoxidized components.

## 5. Conclusions

Tea geometrids (*E. obliqua*) are important pests at Chinese tea plantations, and the sex pheromone of this species is a blend of *Z*3,*Z*6,*Z*9-18:H, *Z*3,epo6,*Z*9-18:H, and *Z*3,epo6,*Z*9-19:H, all of which are typical Type-II sex pheromones. In the present study, we used a combination of comparative RNAseq, tissue expression profile analyses, and phylogenetic relationship analyses to screen for putative candidate genes involved in tea geometrid sex pheromone biosynthesis. The results revealed that two elongases (ELO3 and ELO5), two reductases (FAR2 and FAR9), one decarbonylase (CYP4G173), and one lipophorin (LIP1) exhibited abdomen-based or -specific expression, suggesting that these genes may be involved in synthesizing the unsaturated polytriene sex pheromone components of the *E. obliqua* sex pheromone in oenocytes. In the pheromone gland, CYP340BD1 is suggested to be the most promising epoxidase for producing the two epoxide components. The identified candidates provide a scientific basis for further functional elucidation of Type-II sex pheromone biosynthesis-related enzymes, which will broaden our knowledge of moth sex pheromone biosynthesis and evolution.

## Figures and Tables

**Figure 1 insects-15-00276-f001:**
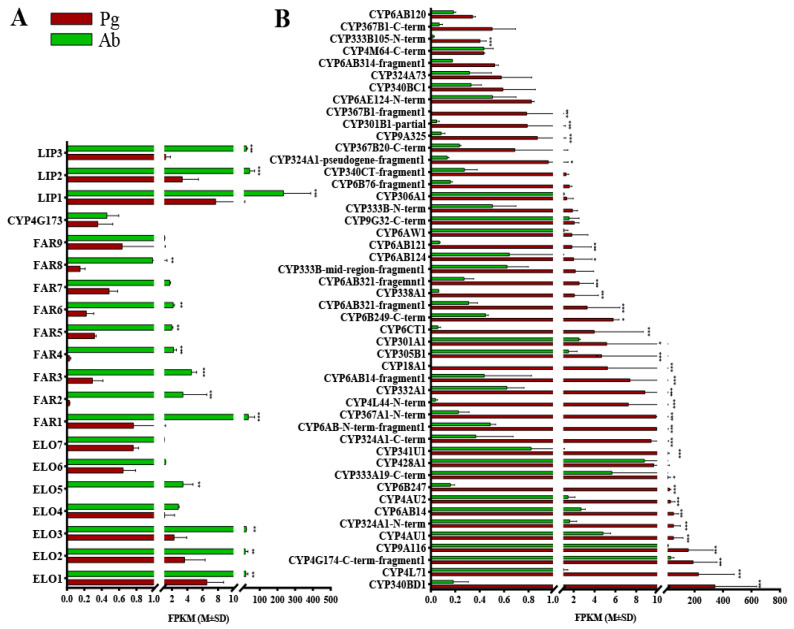
Expression levels of the candidate genes in the tea geometrid (*E. obliqua*) based on the FPKM values, according to comparative RNAseq analyses. (**A**) ELO, FAR, DEC, and LIP genes with higher FPKM values in the abdomen. (**B**) P450s with higher FPKM values in the pheromone gland. Pg and Ab indicate the pheromone gland and the abdomen, respectively. Asterisks represent significant differences between the two tissues based on the Bonferroni *t*-test (* *p* < 0.05, ** *p*< 0.05, *** *p* < 0.001).

**Figure 2 insects-15-00276-f002:**
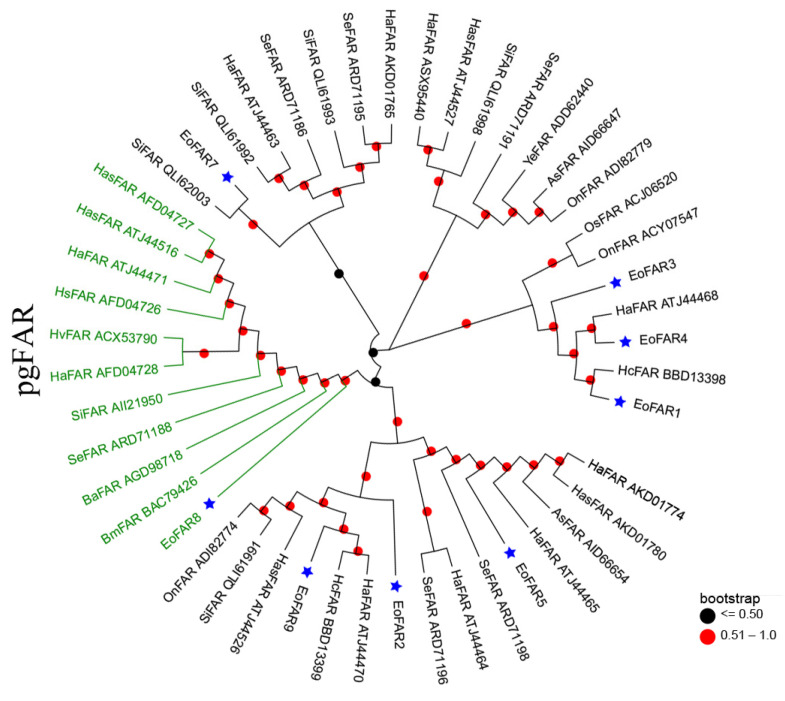
Phylogenetic tree of eight *E. obliqua* FARs predominantly expressed in the abdomen and those from different lepidopteran species. The FAR sequences were retrieved from GenBank and the EST database using BLAST searches with EoFARs as queries. The tree was constructed with MEGA 7.0 based on the amino acid sequences using the neighbor-joining algorithm with the p-distance model. A total of 1000 bootstrap replicates were used. FARs specific to PGs are highlighted in green, and the blue stars indicate FARs from *E. obliqua*. Species abbreviations are as follows: Eo, *E. obliqua*; Ha, *Helicoverpa armigera*; Se, *Spodoptera exigua*; Hc, *Hyphantria cunea*; Has, *Helicoverpa assulta*; Si, *Streltzoviella insularis*; Ba, *Bicyclus anynana*; Hs, *Heliothis subflexa*; As, *Agrotis segetum*; Hv, *Heliothis virescens*; On, *Ostrinia nubilalis*; Os, *Ostrinia scapulalis*; Ye, *Yponomeuta evonymella*.

**Figure 3 insects-15-00276-f003:**
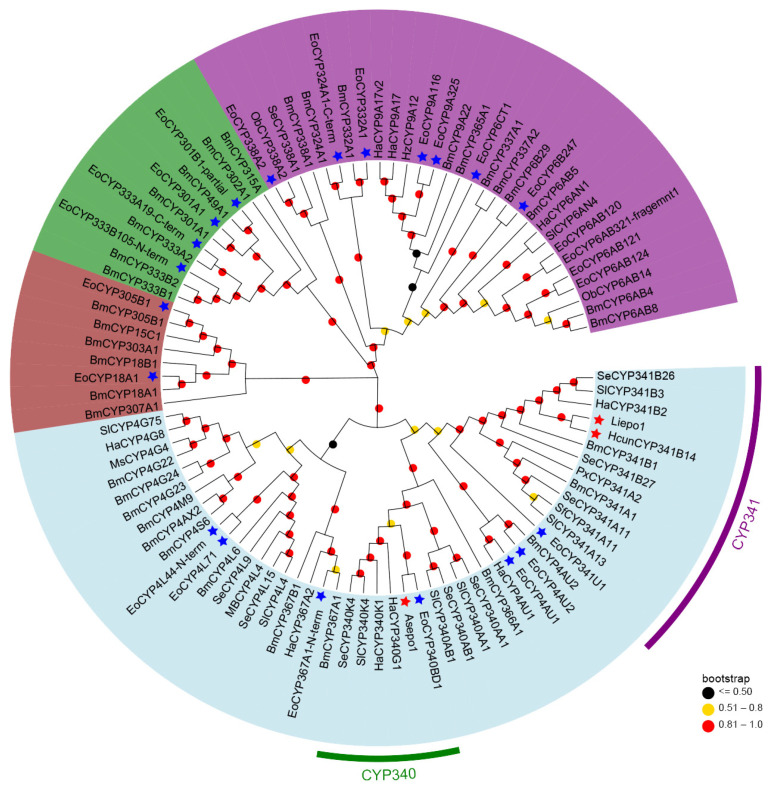
Phylogenetic analysis of 24 *E. obliqua* CYPs predominantly expressed in pheromone glands and those from other insect species. Full-length protein sequences of CYPs from *E. obliqua* and other insects were aligned using ClustalW. The tree was constructed with MEGA 7.0 using the maximum-likelihood method based on the JTT model and tested with a bootstrap value of 1000. Dots with different colors at each branch point represent the bootstrap values. Different background colors indicate the CYP2 clan (dark brown), CYP3 clan (purple), CYP4 clan (light blue), and mitochondrial CYP clan (green). The CYPs marked with blue stars indicate EoCYPs, while the CYPs marked with red stars indicate the 3,4-epoxidase identified in the CYP340 family and the 9,10-epoxidases in the CYP341 family. Species abbreviations are as follows: Eo, *E. obliqua*; Bm, *Bombyx mori*; Ha, *Helicoverpa armigera*; Hz, *Helicoverpa zea*; Sl, *Spodoptera littoralis*; Se, *Spodoptera exigua*; Px, *plutella xylostella*; Ob, *Operophtera brumata*; Ms, *Manduca sexta*; Sf, *Spodoptera frugiperda*.

**Figure 4 insects-15-00276-f004:**
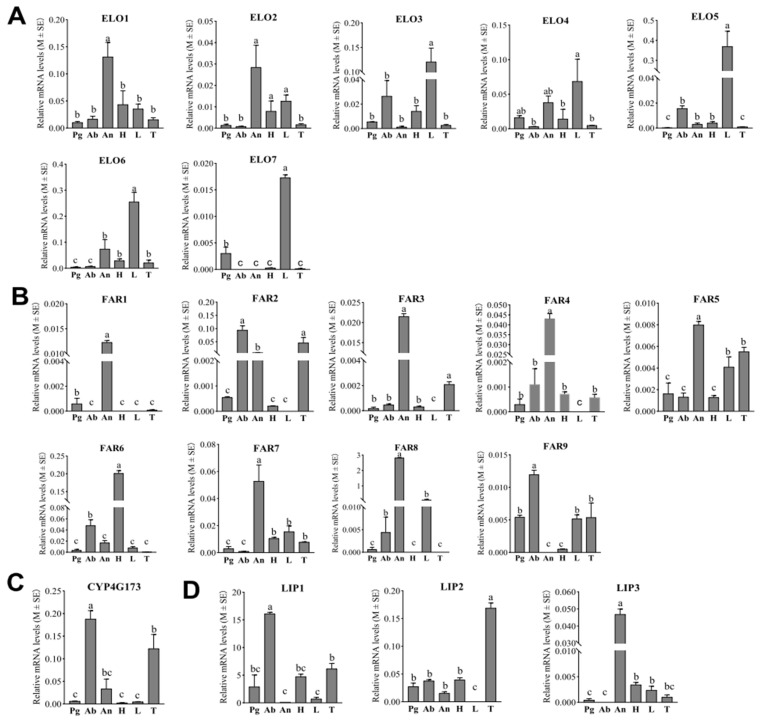
Tissue expression patterns of candidate ELOs, FARs, DECs and LIPs in *E. obliqua* based on RT-qPCR analyses. (**A**–**D**) indicates expression pattern of ELOs; FARs; DEC and LIPs, respectively. Data are shown as means ± standard deviations. Different letters above the bars represent significant differences (*p* < 0.05, one-way ANOVA with a post hoc Duncan’s test). Pg, pheromone gland; Ab, abdomen; An, antennae; H, head; L, legs; T, thorax.

**Figure 5 insects-15-00276-f005:**
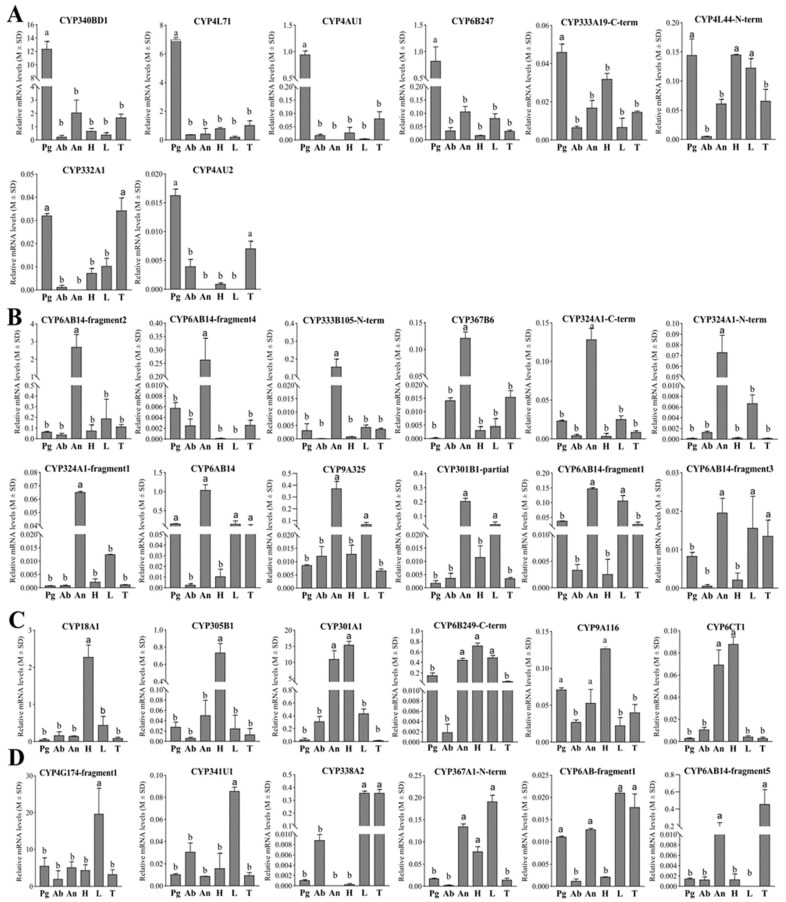
Tissue expression profiles of the 32 preselected pheromone-gland-enriched CYPs in *E. obliqua* based on qPCR analyses. (**A**) The 8 CYPs with upregulated expression in pheromone glands. (**B**) The 12 CYPs with upregulated expression in antennae. (**C**) The 6 CYPs with upregulated expression in head. (**D**) The 5 CYPs with upregulated expression in legs and the 1 CYP with upregulated expression in the thorax. Abbreviations: Pg, pheromone gland; Ab, abdomen; An, antennae; H, head; L, leg; T, thorax. Different letters indicate that the expression levels are significantly different (*p* < 0.05).

**Table 1 insects-15-00276-t001:** Summary of the statistics of the *E. obliqua* transcriptome analyses.

Statistics	Data
Total unigene number	46,202
Total unigene length	48,016,811
Total transcript number	88,962
Total transcript length	110,465,377
Average length	1039
Largest unigene	28,733
NR-annotated unigenes	17,683
GO-annotated unigenes	7499
COG-annotated unigenes	13,283
KEGG-annotated unigenes	7499
SWSS-annotated unigenes	9435

## Data Availability

The original data presented in this study are included in this article or the Appendix A. Further inquiries can be directed to the corresponding authors.

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
