# Peer review of "Identification of Candidate Genes Associated with Type-II Sex Pheromone Biosynthesis in the Tea Geometrid (Ectropis obliqua) (Lepidoptera: Geometridae)"

_insects, 2024, doi:10.3390/insects15040276_

Round 1

Reviewer 1 Report (New Reviewer)

Comments and Suggestions for Authors

The manuscript by Xu et al describes in silico identification and expression profiling of genes potentially involved in the biosynthesis of type II sex pheromones in Ectropis obliqua. Using pheromone gland and abdominal tip specific transcriptomic datasets generated as part of the study, the authors identified putative homologs of lepidopteran sex pheromone biosynthetic pathways (fatty acyl-CoA elongases, fatty acyl-CoA reductases, decarboxylases, lipophorins, and P450s) based on sequence similarity with query sequences. Differential transcript abundance among the datasets and phylogenetic analyses were used as initial screens to identify genes of interest. Subsequent qRT-PCR analyses further refined the dataset of potential biosynthetic pathway genes.

While the manuscript is essentially a descriptive data mining study, it is well-constructed and represents an important first step in elucidating the individual components of the Ectropis obliqua sex pheromone biosynthetic pathway. These findings will be crucial for assessing the efficacy that targeted disruption of pheromone biosynthesis may have for management of this pest species and will undoubtedly provide critical insights into the biosynthesis of type II sex pheromones in general.  The manuscript, however, could be strengthened by more thorough editing, better clarification of citations, and the inclusion of key methodological details. In addition, the authors might consider including BUSCO metrics as a measure of the completeness of the transcriptomic datasets mined. Specific comments/questions are listed below.

Specific comments

1) More thorough English editing either via a scientific editing service or a native English-speaking colleague with experience in the field would make the paper more accessible. As currently written, the sentence structure and use of verb tenses make it difficult to determine what was done previously, what the authors are citing, and what was done in the current study.

2) It is difficult to differentiate hypotheses regarding pathway mechanisms and sections of the manuscript that are referencing previous work without obvious citations. Further, previous research and specific references to software and results have not been appropriately cited. Examples include:

- lines 96-104 - Are these hypothesized findings or have the steps been demonstrated? If the later, then citations should be included.

- line 167 - Citations should be provided for the genes published previously. Also, if the protein sequences were used as queries to search the respective transcriptomes, then a supplemental table listing accession numbers of the proteins used should be provided.

- line 181 - A citation should be provided for MEGA 7.

- line 200 – Is there a reference for the stability of the genes used in qPCR normalization?

- lines 236-238 – Is this speculation or has it been shown? If shown, then the appropriate references should be cited.

- lines 251-254 – Is this speculation or has it been shown? If shown, then the appropriate references should be cited.

- line 267 – is the lepidopteran pgFAR clade a defined clade? Is there a reference for this?

- line 287 –References for the previously identified epoxidases should be cited.

- lines 318-319 – Is this speculation or has it been shown? If shown, then the appropriate references should be cited.

- lines 427-428– Is this speculation or has it been shown? If shown, then the appropriate references should be cited.

3) Include more methodological details

- line 148 - What version of Trinity was used? What parameters were used? Trinity should be cited.

- line 158 - What program was used to assess expression levels?

- line 180 - Were the alignments done with full-length proteins? Fragments? Defined protein domains? Based on the Results, it seems that some of the putative Ectropis sequences were excluded based on size considerations. This should be described in more detail here. Also, how were gaps/deletions handled in the phylogenetic analysis?

- line 202 – Was the specificity of the qPCR confirmed with melt curve analyses? Were the qPCR products sequence validated? Were primer efficiencies considered? If so, how were they determined and what were the values?

- line 438 – The Bombyx pgFAR is arguably the first FAR functionally characterized and should be cited accordingly.

4) BUSCO metrics describing the completeness of the respective assemblies would be helpful for evaluating the utility of the two transcriptomic datasets. Please see https://busco.ezlab.org/

5) Lines 379-380  reference CYPs notably expressed in antennae (Fig 5B), heads (Fig 5C), and legs (Fig5D); however, when looking at Figure 5 it seems that some of the CYPs grouped within these designations also exhibit elevated expression in other tissues that is not statistically different from the listed tissue (Eg CYP6AB14 is elevated in Pg, An, L, and T, yet is clustered with the “antennae-enriched” graphs). Either split the figure into additional sections that more accurately reflect the text or modify the text. Also, were any of the CYP clades (or representatives of those clades) described in Figure 3 predominantly expressed in a single tissue?

6) Line 354 states that the qPCR expression profile of CYP4G173 was “inconsistent with the RNA-Seq results”. Please elaborate and provide a reason for why this discrepancy.

Minor comments

- line 22 - clarify that these are type II sex pheromone blends

- line 56 - delete or

- line 60 - punctuation after [5] is confusing  

- lines 251-260 – should these lines be italicized?

- line 265 – “…FAR7 established its own distinct clade”. This wording implies that FAR7 is the only member of the clade. Rather, it forms a unique clade with SiFAR QLI62003.

- line 271- lepidopteran

- line 309 – Spodoptera littoralis should be italicized

- line 310 – Mb should be Ms

- For better clarity, all of the Figures could be enlarged

Comments on the Quality of English Language

More thorough English editing either via a scientific editing service or a native English-speaking colleague with experience in the field would make the paper more accessible. As currently written, the sentence structure and use of verb tenses make it difficult to determine what was done previously, what the authors are citing, and what was done in the current study.

Author Response

Thank you very much for your fruitful and constructive suggestions. We have considered them thoroughly and revised our manuscript accordingly.

Specific comments

1) More thorough English editing either via a scientific editing service or a native English-speaking colleague with experience in the field would make the paper more accessible. As currently written, the sentence structure and use of verb tenses make it difficult to determine what was done previously, what the authors are citing, and what was done in the current study.

Answer: To address our results and conclusion more clear, we have edited the language throughout the full text.

2) It is difficult to differentiate hypotheses regarding pathway mechanisms and sections of the manuscript that are referencing previous work without obvious citations. Further, previous research and specific references to software and results have not been appropriately cited. Examples include:

- lines 96-104 - Are these hypothesized findings or have the steps been demonstrated? If the later, then citations should be included.

Answer: These are hypothesis which based on previous reports from other moths also using the same compound 3Z,6Z,9Z-18:H as its sex pheromone component. We have changed the verb tenses to present tenses.

- line 167 - Citations should be provided for the genes published previously. Also, if the protein sequences were used as queries to search the respective transcriptomes, then a supplemental table listing accession numbers of the proteins used should be provided.

Answer: Here we didn’t use specific protein sequences are queries to search for the putative biosynthetic enzyme in E. obliqua. Instead, we searched the enzymes’ name in the annotation tables, and then valid all the retrieved sequence in NCBI. To avoid the misleading description in the previous version, we have corrected the text to the follows in the revised manuscript.

The putative enzymes linked to E. obliqua sex pheromone biosynthesis and transportation (namely ELO, FAR, DEC, LIP, and P450 epoxidase) were identified from the NR annotation results by searching the enzymes’ name. The previous sentence “Based on the moth sex pheromone biosynthesis genes published previously” has been removed.

- line 181 - A citation should be provided for MEGA 7.

Answer: The related reference about MEGA 7 has been cited.

- line 200 – Is there a reference for the stability of the genes used in qPCR normalization?

Answer: The related reference has been cited.

- lines 236-238 – Is this speculation or has it been shown? If shown, then the appropriate references should be cited.

Answer: This is speculation based on previous reports from other moths also using the same compound 3Z,6Z,9Z-18:H as its sex pheromone component. We have corrected the description to the follows.

Originated from linolenic acid, the biosynthesis of the polyene hydrocarbons sex pheromone components in E. obliqua is supposed to proceed with consecutive reactions including chain elongation, oxidation, reduction and decarbonylation within the oenocytes of the abdomen.

- lines 251-254 – Is this speculation or has it been shown? If shown, then the appropriate references should be cited.

Answer: This has been shown. And the related references have been cited.

- line 267 – is the lepidopteran pgFAR clade a defined clade? Is there a reference for this?

Answer: The related reference has been cited, please see line 443-444.

- line 287 –References for the previously identified epoxidases should be cited.

Answer: Information about the previously identified epoxidases has been citated in the introduction. Line 287 was written in the result analysis part to explain why we focused on the CYPs within clan4. So we didn’t add the citations here.

- lines 318-319 – Is this speculation or has it been shown? If shown, then the appropriate references should be cited.

Answer: These are hypothesis which based on previous reports. We have changed the verb tenses to present tenses.

- lines 427-428– Is this speculation or has it been shown? If shown, then the appropriate references should be cited.

Answer: Compared to FARs, attentions paid to ELOs, DECs and LIPs are relatively scarce. Most ELOs and DECs (CYP4Gs) associated with cuticular hydrocarbon biosynthesis have been well-characterized [32].

The form sentence is my opinion, because when searching the literature, we have gotten lots of report about FARs, reports about ELOs, DECs and LIPs are relative less. The latter sentence, we have added the citation.

3) Include more methodological details

- line 148 - What version of Trinity was used? What parameters were used? Trinity should be cited.

Answer: Trinity version 2.15.1 (https://github.com/trinityrnaseq/trinityrnaseq/releases) was used for analysis with default settings. We have added this information and the citation in the text accordingly

- line 158 - What program was used to assess expression levels?

Answer: The expression levels of identified transcripts were calculated with the software RSEM v1.3.3 (http://deweylab.github.io/RSEM/). We have added this information in the text accordingly.

- line 180 - Were the alignments done with full-length proteins? Fragments? Defined protein domains? Based on the Results, it seems that some of the putative Ectropis sequences were excluded based on size considerations. This should be described in more detail here. Also, how were gaps/deletions handled in the phylogenetic analysis?

Answer: The alignments were done with both full-length proteins sequence and fragments using ClustalW implemented in MEGA 7.0 with default setting. No specific protein domains were defined. For the phylogenetic analysis of FARs, the gaps/deletions were also used with the default setting of the MEGA 7.O. We have changed the text in the manuscript accordingly to the follows.

Specifically, the tree for FARs was generated using the neighbor-joining method, incorporating the - p-distance model, while the maximum likelihood method based on the Jones-Taylor-Thornton (JTT)  model was employed for CYP phylogenetic analysis. For both analyses, a bootstrap value of 1,000 was used, and all the other parameters were default.    

- line 202 – Was the specificity of the qPCR confirmed with melt curve analyses? Were the qPCR products sequence validated? Were primer efficiencies considered? If so, how were they determined and what were the values?

Answer: Yes, we checked the primer specificity by conducting a melt curve analysis using Roche LightCycler 480. Only the primers that exhibited a single peak were selected and used for further qPCR analysis. As the primer specificity has been confirmed by melting curve analysis, we didn’t validate the qPCR products by further sequencing. The primer efficiencies were evaluated according to the standard curve method mentioned in the citated reference “Standardization of qPCR and RT-qPCR”. Since only primers with amplification efficiencies between 90 -110% can be used for further analysis according to the MIQE guidelines. We didn’t present each amplification efficiencies data in the text.

- line 438 – The Bombyx pgFAR is arguably the first FAR functionally characterized and should be cited accordingly.

Answer: The Bombyx pgFAR has been cited.

4) BUSCO metrics describing the completeness of the respective assemblies would be helpful for evaluating the utility of the two transcriptomic datasets. Please see https://busco.ezlab.org/

Answer: Thanks for the suggestion. For the present manuscript, we have used Trimmomatic version 0.40 for trimming and quality control, and Trinity version 2.15.1 for sequence assemble, which includes the examination of the completeness of an assembly. Anyway, thank you for the introduction of BUSCO, we will use this software for our next project.

5) Lines 379-380  reference CYPs notably expressed in antennae (Fig 5B), heads (Fig 5C), and legs (Fig5D); however, when looking at Figure 5 it seems that some of the CYPs grouped within these designations also exhibit elevated expression in other tissues that is not statistically different from the listed tissue (Eg CYP6AB14 is elevated in Pg, An, L, and T, yet is clustered with the “antennae-enriched” graphs). Either split the figure into additional sections that more accurately reflect the text or modify the text. Also, were any of the CYP clades (or representatives of those clades) described in Figure 3 predominantly expressed in a single tissue?

Answer: Thanks for the suggestion. We have changed the text to the follows.

Furthermore, among the 32 preselected CYPs, 7 CYPs demonstrated a pronounced accumulated expression pattern specifically in the antennae, and 5 additional CYPs exhibited also antennae-preferentially expression pattern (Figure 5B), 6 CYPs were preferentially expressed in the head, moreover, a significant high level in both CYP18A1 and CYP305B1 expression were observed in the head(Figure 5C). In addition, 5 CYPs were detected to express preferentially in the legs, and CYP4G174-fragment1 and CYP341U1 were found to be significantly up-regulated in the legs (Figure. 5D).

For the second question, in the Figure3. CYP340BD1 and CYP341U1 were clustered with the known epoxidases into CYP340 and CYP341 families, respectively. And these two genes were detected to predominantly expressed only in pheromone gland and legs, respectively. (Figure 5A and Figure 5D).

6) Line 354 states that the qPCR expression profile of CYP4G173 was “inconsistent with the RNA-Seq results”. Please elaborate and provide a reason for why this discrepancy.

Answer: For qPCR analysis, we analyzed the expressional level of CYP4G173 in pheromone gland, abdomen, antennae, head, legs and thorax. The results showed CYP4G173 exhibited significant high expression in the abdomen of female E. obliqua compared to other tissues (Figure. 4C). RNAseq results also showed that CYP4G173 expressed more in the abdomen than in the pheromone glands. Therefore, the qPCR result is in consistent with the RNAseq results. There is no discrepancy.

Minor comments

- line 22 - clarify that these are type II sex pheromone blends

Answer: we have added the word sex in the sentence.

- line 56 - delete or

Answer: Has been deleted.

- line 60 - punctuation after [5] is confusing  

Answer: we have deleted unnecessary punctuations.

- lines 251-260 – should these lines be italicized?

Answer: Thanks for pointing out the mistakes, we have corrected the mistakes.

- line 265 – “…FAR7 established its own distinct clade”. This wording implies that FAR7 is the only member of the clade. Rather, it forms a unique clade with SiFAR QLI62003.

Answer: we have changed the misleading description, now the sentence is as follows.

while FAR7 established a distinct clade with SiFAR QLI62003

- line 271- lepidopteran

Answer: corrected.

- line 309 – Spodoptera littoralis should be italicized

Answer: corrected.

- line 310 – Mb should be Ms

Answer: corrected.

- For better clarity, all of the Figures could be enlarged

Answer: We have enlarged all the figures.

Reviewer 2 Report (Previous Reviewer 1)

Comments and Suggestions for Authors

The authors have addressed all my comments, and the paper has been significantly improved after revising.

Author Response

Dear reviewer,

Thank you very much for your valuable suggestions.

Best regards.

Round 2

Reviewer 1 Report (New Reviewer)

Comments and Suggestions for Authors

The authors' considerations for the reviewer comments are appreciated. My suggestions/questions have been sufficiently addressed. My only comment, which likely could be addressed during the proofs stage of publication, is that reference 32 (Qiu et al., 2012) is a duplicate of reference 26 (Qui et al., 2012).

Author Response

The authors' considerations for the reviewer comments are appreciated. My suggestions/questions have been sufficiently addressed. My only comment, which likely could be addressed during the proofs stage of publication, is that reference 32 (Qiu et al., 2012) is a duplicate of reference 26 (Qui et al., 2012).

Answer: Thanks for pointing out the mistake. We have deleted the duplicate. 

This manuscript is a resubmission of an earlier submission. The following is a list of the peer review reports and author responses from that submission.

Round 1

Reviewer 1 Report

Comments and Suggestions for Authors

In the manuscript entitled “Identification of Genes Associated with Type-II Sex Pheromone Biosynthesis in the Tea Geometrid, Ectropis obliqua (Lepidoptera: Geometridae)”, Xu and colleagues used RNA-seq to analyze the female E. obliqua pheromone gland and the abdomen transcriptomes to ascertain the genes related to the biosynthesis of its sex pheromone components. Furthermore, using quantitative PCR, they confirm the biased expression genes in pheromone gland and the abdomen. The results will provide insight into the genes involved in the biosynthesis of sex pheromone components and will contribute to our understanding of the moth sex pheromone evolution.

The manuscript is written clearly and concisely, and the analyses are adequate to draw the conclusions presented. I have only some minor suggestions to the manuscript.

Figure 2: please provide the scale of the color legend. In addition, please indicate whether the FPKM values have been normalized. The volcano plot is suggested instead of the heatmap in here.

In table 1, what does the “largest unigene” means?

Page 3 line 108: change the “SWISS” to “SWISS-Prot”.

Page 3 line 115-116: “... the most highly expressed genes ...”, please check that these genes highly expressed in which samples?

Page 4 line 124: what does the “in” means?

Page 5 line 138: a fold change describes the ratio of two values, what does these two values refer to here?

Comments on the Quality of English Language

The manuscript is written clearly and concisely, refer to the comments and suggestions for authors.

Author Response

Comments and Suggestions for Authors

In the manuscript entitled “Identification of Genes Associated with Type-II Sex Pheromone Biosynthesis in the Tea Geometrid, Ectropis obliqua (Lepidoptera: Geometridae)”, Xu and colleagues used RNA-seq to analyze the female E. obliqua pheromone gland and the abdomen transcriptomes to ascertain the genes related to the biosynthesis of its sex pheromone components. Furthermore, using quantitative PCR, they confirm the biased expression genes in pheromone gland and the abdomen. The results will provide insight into the genes involved in the biosynthesis of sex pheromone components and will contribute to our understanding of the moth sex pheromone evolution.

The manuscript is written clearly and concisely, and the analyses are adequate to draw the conclusions presented. I have only some minor suggestions to the manuscript.

Figure 2: please provide the scale of the color legend. In addition, please indicate whether the FPKM values have been normalized. The volcano plot is suggested instead of the heatmap in here.

Answer: To plot the original figure 2 in TBtool, the average FPKM of each gene was transformed using log base 2 calculator, and was further normalized between 0 to 1. And the color legend of figure 2 was shown in the right corner, and in the figure legend, we have also added the related information for FPKM normalization and figure plot details. A volcano plot is possible here, but all the genes will be presented as dots in the figure, then it will hard to observe whether certain gene is upregulated or downregulated. Taking the suggestion into consideration, in the revised version, we plotted a new bar figure of the candidate genes in the tea geometrid E. obliqua based on the FPKM values according to comparative RNAseq analysis., see figure 1.

In table 1, what does the “largest unigene” means?

Answer: After transcriptome assemble, we got a pool of unigenes with different lengths. Among them, the unigene with the longest nucleotides is defined as the largest unigene. 

Page 3 line 108: change the “SWISS” to “SWISS-Prot”.

Answer: SWSS is the abbreviation of “SWISS-Prot”, we have added this abbreviation at where SWISS-Prot arose for the first time in the text. And the SWSS abbreviation was used thereafter.

Page 3 line 115-116: “... the most highly expressed genes ...”, please check that these genes highly expressed in which samples?

Answer: Thanks for pointing out the mistakes. Here we intended to state that the unigenes were assigned to 322 different pathways, and among the 322 pathways. the metabolic pathways and the biosynthesis of secondary metabolites were the top2 dominant pathway, accounting for 1256 and 433 unigenes, respectively. And we have changed the sentence as follows.

In addition, KEGG analysis revealed that the unigenes were assign to 322 different pathways, where metabolic pathways were dominant (1256 unigenes), followed by the biosynthesis of secondary metabolites, involving 433 unigenes.

Page 4 line 124: what does the “in” means?

Answer: the “in” has been deleted.

Page 5 line 138: a fold change describes the ratio of two values, what does these two values refer to here?

Answer: Yes, here the description was wrong, we have added the detail data processing procedures and have changed the description accordingly to the follows.

The average FPKM of each gene was transformed using log base 2 calculator, and was further normalized between 0 to 1 using TBtool. The color indicates the normalized expressional level of each gene.

Reviewer 2 Report

Comments and Suggestions for Authors

The paper entitled “Identification of Genes Associated with Type-II Sex Phero-2 mone Biosynthesis in the Tea Geometrid, Ectropis obliqua 3 (Lepidoptera: Geometridae)” describes the transcriptome analysis of female tea moths, Ectropis obliqua. The authors were looking for putative genes that encode enzymes involved in the biosynthesis of the moth’s pheromone, a mixture of cis epoxides with one or two Z double bonds.

The results obtained include enzymes involved in fatty acyl CoA reduction (to the aldehyde), several cytochromes P450 (from clades that could function as the epoxidase of hydrocarbon alkene precursors, as well as from the CYP4G group that is known to be involved in hydrocarbon biosynthesis in insects).  They also looked for and found some fatty acyl CoA elongases and lipophorin.

Major items

1) The authors make reference to alpha oxidation of Z11,14,17-20:SCoA to Z10,13,16 19:SCoA.  As written that process is not possible.  It would require a multi-step process which includes alpha oxidation, possibly to the ketone, hydrolysis of the CoA ester to the carboxylic acid, then decarboxylation of the alpha-keto acid. To identify the enzyme that trims the Fatty acyl CoA by one carbon, they would need to look for: a hydrolase, an alpha oxidizing enzyme and the decarboxylase.  For this type of reaction, they need to look for an enzyme that uses thyamine pyrophosphate on an alpha keto acid. Two possible products could result from alpha “trimming”, depending on the enzyme: either the aldehyde or the acyl CoA one C shorter than the original fatty acid. The latter would require lipoic acid, FAD and CoA as additional cofactors.

2) What evidence is there that the hydrocarbon precursors specifically needed for pheromone biosynthesis are produced in the oenocytes and not in the pheromone gland itself?

3) Given that the pheromone gland is between the 9th and 10th abdominal segments, how are you separating it from fat body tissues?  There is no indication of dissection procedures used in the methods.

4) The paper is written in a way (tense) that is misleading.  It gives the impression that the genes found are the ones mapping the biosynthetic pathway of the pheromone in this moth, which is not correct.  Only some genes have been found to be expressed in various tissues, and it is not clear how well those tissues were separated from each other (see 3 above).

5) Additional experiments should be done to support the claims made of involvement of the genes that encode putative enzymes in the pathway of pheromone biosynthesis.

First, the authors could do in situ hybridization, coupled with microscopy, to track the expression of key genes of interest in, for example, the pheromone gland or the fat body.

Second, it should be relatively easy to feed the moths 1-13C labeled linoleic and linolenic acids, then analyze the pheromone produced for labeling. If the pathway they propose is correct, the 1-13C label should be incorporated into the pheromone.

Minor items

Line 43.  “bondsalongside” (space missing)

Line 51. “…majority of… (add “the” in front of “majority”)

Line 118 E. obliqua not E. oblique

Fig 1: left side. The conversion of the fatty aldehyde to the hydrocarbon is a decarbonylation, not a decarboxylation)

Lines 121-126 and lines 154-164: wrong tense.  Please see above, comment 4.

Comments on the Quality of English Language

See above.

Author Response

Comments and Suggestions for Authors

The paper entitled “Identification of Genes Associated with Type-II Sex Phero-2 mone Biosynthesis in the Tea Geometrid, Ectropis obliqua 3 (Lepidoptera: Geometridae)” describes the transcriptome analysis of female tea moths, Ectropis obliqua. The authors were looking for putative genes that encode enzymes involved in the biosynthesis of the moth’s pheromone, a mixture of cis epoxides with one or two Z double bonds.

The results obtained include enzymes involved in fatty acyl CoA reduction (to the aldehyde), several cytochromes P450 (from clades that could function as the epoxidase of hydrocarbon alkene precursors, as well as from the CYP4G group that is known to be involved in hydrocarbon biosynthesis in insects).  They also looked for and found some fatty acyl CoA elongases and lipophorin.

Major items

1) The authors make reference to alpha oxidation of Z11,14,17-20:SCoA to Z10,13,16 19:SCoA.  As written that process is not possible.  It would require a multi-step process which includes alpha oxidation, possibly to the ketone, hydrolysis of the CoA ester to the carboxylic acid, then decarboxylation of the alpha-keto acid. To identify the enzyme that trims the Fatty acyl CoA by one carbon, they would need to look for: a hydrolase, an alpha oxidizing enzyme and the decarboxylase.  For this type of reaction, they need to look for an enzyme that uses thyamine pyrophosphate on an alpha keto acid. Two possible products could result from alpha “trimming”, depending on the enzyme: either the aldehyde or the acyl CoA one C shorter than the original fatty acid. The latter would require lipoic acid, FAD and CoA as additional cofactors.

Answer: Thanks very much for the constructive suggestions. In reference 10, Groll et al. proved that [2,2,3,3-2H4]-10Z,13Z,16Z-19:COOH and [3,3,4,4-2H4]-11Z,14Z,17Z-20:COOH were incorporated to the sex pheromone component Z3, Z6, Z9-18:H. Therefore, he proposed that Z10,13,16 19:SCoA was derived from Z11,14,17-20:SCoA via alpha oxidation. To the best of our knowledge, although α -oxidation of myristic acid has been shown to catalyzed by a P450-like enzymes in the bacterium Sphingomonas paucimobilis ( reference 36), α-oxidation of unbranched fatty acids in insects has not been reported before. As a results, in our manuscript, we didn’t looked for enzymes that are potentially involved in the alpha oxidation of Z11,14,17-20:SCoA to Z10,13,16 19:SCoA. Of course, as you have suggested, multiple enzymes are probably required for this process. In either cases, the production of Z10,13,16 19:SCoA in the tea geometrid are interesting and of vital importance for the biosynthesis of Z3, Z6, Z9-18:H. More experiments are needed to fully solved this puzzle. 

2) What evidence is there that the hydrocarbon precursors specifically needed for pheromone biosynthesis are produced in the oenocytes and not in the pheromone gland itself?

Answer: According to article “Transport of a hydrophobic biosynthetic precursor by lipophorin in the hemolymph of a geometrid female moth which secretes an epoxyalkenyl sex pheromone” (see reference 14, DOI: 10.1016/j.ibmb.2006.04.006), we got to know that only epoxidation proceeds in the pheromone gland, while the polyunsaturated hydrocarbons are probably produced in the oenocytes attached to either epidermal cells or fat body cells of the abdomen.

3) Given that the pheromone gland is between the 9th and 10th abdominal segments, how are you separating it from fat body tissues?  There is no indication of dissection procedures used in the methods.

Answer: To collect the rod-shaped pheromone gland, the terminal parts of the abdomen of virgin female (2 – 3 days old) was squeezed gently. The extruded pheromone gland was excised with fine scissors. We have added this information in the methods. 

4) The paper is written in a way (tense) that is misleading.  It gives the impression that the genes found are the ones mapping the biosynthetic pathway of the pheromone in this moth, which is not correct.  Only some genes have been found to be expressed in various tissues, and it is not clear how well those tissues were separated from each other (see 3 above).

Answer: More detailed information regarding to the dissection of pheromone gland and abdomen was added in the methods part, see 3 as well. For the other tissues, like head, legs, thorax and antennae, they were collected by directly cutting laterally at the junctions. For the tense, we have changed the misleading parts. In the manuscript, we presented the enzymes (or genes) results in their order of occurrence in the biosynthetic pathway, namely, elongation, α-oxidation, reduction, and decarbonylation, transportation and epoxidation.

5) Additional experiments should be done to support the claims made of involvement of the genes that encode putative enzymes in the pathway of pheromone biosynthesis.

First, the authors could do in situ hybridization, coupled with microscopy, to track the expression of key genes of interest in, for example, the pheromone gland or the fat body.

Second, it should be relatively easy to feed the moths 1-13C labeled linoleic and linolenic acids, then analyze the pheromone produced for labeling. If the pathway they propose is correct, the 1-13C label should be incorporated into the pheromone.

Answer: Wonderful suggestions. We admitted that functional characterization of these candidate genes is of vital importance. We are currently working on the characterization of epoxidase based on the candidates screened in this manuscript. Hopefully, we could report the related results soon.

Regarding to the biosynthetic pathway, in the winter moth, Erannis bajaria, Wang et al. and Ding et al. have been proven that deuterium labeled linolenic acid, (D4-Z10,Z13,Z16-19:Acid) and (11Z,14Z,17Z)-[2,2,3,3,4,4-2H6 ]-11,14,17-icosatrienoic acid (D6-Z11,Z14,Z17-20:Acid) were incorporated to the sex pheromone component Z3, Z6, Z9-19:H and 1,Z3,Z6,Z9-19:H. Details can be found in reference 10 and 11. Moreover, Groll et al. proved that [2,2,3,3-2H4]-10Z,13Z,16Z-19:COOH and [3,3,4,4-2H4]-11Z,14Z,17Z-20:COOH were incorporated to the sex pheromone component Z3, Z6, Z9-18:H, see reference 6. The Z11,Z14,Z17-20:Acid was the malonate elongation product of linolenic acid, and 10Z,13Z,16Z-19:COOH was derived from Z11,Z14,Z17-20:Acid. Since the tea geometrid Ectropis obliqua also produces Z3, Z6, Z9-18:H and Z3, Z6, Z9-19:H, we speculated the biosynthetic pathway is the same to the winter moth. To avoid the confusion, we have rewrote the introduction parts, to make this background information more clear. Of course, you are absolute right, experiments results are the most direct proofs. We will conduct labeling precursor feeding or injection assays later.

Minor items

Line 43.  “bondsalongside” (space missing)

Answer: Thank you for your correction, “bondsalongside” is changed as “bonds alongside”

Line 51. “…majority of… (add “the” in front of “majority”)

Answer: Changed.

Line 118 E. obliqua not E. oblique

Answer: Mistake has been corrected.

Fig 1: left side. The conversion of the fatty aldehyde to the hydrocarbon is a decarbonylation, not a decarboxylation)

Answer: Appreciated for the comment, “decarboxylation” has been changed to “decarbonylation” in Fig.1.

Lines 121-126 and lines 154-164: wrong tense.  Please see above, comment 4.

Answer: changed. 

Reviewer 3 Report

Comments and Suggestions for Authors

This paper describes the result of transcript analysis in the pheromone gland of Tea Geometrid, Ectropis obliqua (Lepidoptera: Geometridae). The results were well analyzed by the authors and conclusion is clear to read.

However, I think this is a report in a database, not a research report. Genes implicated in pheromone biosynthesis are classified into multigene family due to duplication on the genome such as desaturase, epoxygenases and reductases. To prevent a description of non-function as pseudogene copies, I recommend author(s) should also confirm and report about their gene function. If impossible, author(s) should mention the risk (i.e., an existence in the non-function copies) in introduction part. Author(s) should learn more about gene duplication (Nei and Rooney 2005; Rooney [www.pnas.org/cgi/doi/10.1073/pnas.1104355108]).

In addition, "oenocyte" of moths should be used more carefully because the entity is unclear. In reference
#11, actually, Coby Schal and Veeresh Sevala confine as “probably oenocyte”. If author(s) has any knowledge of the “oenocyte”, provide it to readers. If impossible, evidence will be required as to why authors remark “oenocyte” in detail.
When two missing are mentioned in introduction in the next version, it will be suitable for publication.

Comments on the Quality of English Language

N/A

Author Response

This paper describes the result of transcript analysis in the pheromone gland of Tea Geometrid, Ectropis obliqua (Lepidoptera: Geometridae). The results were well analyzed by the authors and conclusion is clear to read.

However, I think this is a report in a database, not a research report. Genes implicated in pheromone biosynthesis are classified into multigene family due to duplication on the genome such as desaturase, epoxygenases and reductases. To prevent a description of non-function as pseudogene copies, I recommend author(s) should also confirm and report about their gene function. If impossible, author(s) should mention the risk (i.e., an existence in the non-function copies) in introduction part. Author(s) should learn more about gene duplication (Nei and Rooney 2005; Rooney [www.pnas.org/cgi/doi/10.1073/pnas.1104355108]).

Answer: Wonderful suggestions. We admitted that functional characterizations, which is underway now, of these genes are of vital importance. We hope we could report the related results soon. As for the existence of pseudogenes, it is possible. And we have mentioned this point in the discussion (line 433-436) in the revised manuscript.

In addition, "oenocyte" of moths should be used more carefully because the entity is unclear. In reference ï¼ƒ11, actually, Coby Schal and Veeresh Sevala confine as “probably oenocyte”. If author(s) has any knowledge of the “oenocyte”, provide it to readers. If impossible, evidence will be required as to why authors remark “oenocyte” in detail.
When two missing are mentioned in introduction in the next version, it will be suitable for publication.

Answer: Thanks very much for the reminder and suggestion. We checked all related reference again, due to the long-chain features and fatty acid origin, the primary enzymatic steps of Type-II sex pheromone biosynthesis proceed similarly to those of insects cuticle hydrocarbon synthesis. And in the mosquitos and drosophila, the isolation of the oenocytes provided direct evidence that oenocytes is the biosynthetic site of hydrocarbons. In terms of the sex pheromone biosynthesis, Ding et al. (reference 7) has identified a fatty-acyl-CoA desaturase involved in the type-II sex pheromone biosynthesis in winter moth. And this enzyme was significantly upregulated in the epidermal cells, which are usually the places where the oenocyte cells associated with. Therefore, presently, it is generally accepted that oenocytes is the biosynthetic site of the polyene hydrocarbons. To address this information more clear, we have added these information in the introduction.